# The Number of Fungiform Papillae, Taste Sensitivity and Smell Functions of Children Aged 11–15

**DOI:** 10.3390/nu14132578

**Published:** 2022-06-22

**Authors:** Grzegorz Sobek, Paweł Jagielski

**Affiliations:** 1Institute of Health Sciences, Medical College of Rzeszów University, 35-959 Rzeszów, Poland; 2Department of Nutrition and Drug Research, Institute of Public Health, Faculty of Health Sciences, Jagiellonian University Medical College, 31-066 Krakow, Poland; paweljan.jagielski@uj.edu.pl

**Keywords:** children, fungiform papillae, taste, smell, sensitivity

## Abstract

Differences in the ability to identify and perceive tastes and smells might influence food consumption and, ultimately, chronic nutrition-related conditions such as overweightness and obesity. This study aimed to investigate the associations between taste sensitivity and odour function, anthropometry, and quantity of fungiform papillae in children at age 11–15. Taste strips (4 base tastes), U-Sniff sticks (12 selected smells), and a filter paper strip impregnated with 6-n-propylthiouracil (PROP) were used. The photographic method was used to estimate the number of fungiform papillae (FP) on the tongue. The results showed that the quantity of FP was not related to anthropometry or gender. The taste test total scores were higher for girls, for whom the median score was 14 (12.0–15.0), than for boys, for whom the median score was 12 (9.0–13.0). Of the children, 13.9% had some difficulty in identifying odours. The Mann–Whitney U test showed that children who were most sensitive to bitter taste had more FP (*p* = 0.0001). The median score for this group (score = 4) was 34.0 (27.0–37.0). For those who had some problems with correctly assessing all bitter taste strips (score = 0–3), the median score was 24.0 (20.0–31.0). Higher numbers of FP were also observed in tasters, that is, people sensitive to PROP, than in nontasters. Only some measures of the taste function correlated with each other, but not very significantly. We concluded that there are multiple perceptual phases of taste, with no single measure able to entirely represent the sense of taste.

## 1. Introduction

The sense organs provide communication with the external environment and decide how we perceive the world around us [1]. Taste is the sensation felt when a substance in the mouth is recognized by the taste receptors of the taste on the fungiform papillae of the tongue. There are five basic tastes: salty, sweet, bitter, sour, and umami (spicy). For some time, it has been proposed to consider the fatty taste as the sixth basic taste [2]. Tasting sensations have been found to influence food choice and food consumption. People differ in the intensity of perceiving different food sensory features, which may influence their food intake and, consequently, their final nutrition choices [3]. As it is known that eating habits are associated with the development of chronic diseases and life quality, the sensory perception of the mouth is a significant aspect to reflect on when developing approaches to preventing these diseases and promoting healthy eating, both among children and adults [4]. The reasons for differences in taste sensitivity are complex and include numerous genetic, physiological, and environmental factors. Other factors influencing sensory perception may include age, sex, nutritional level, diseases, and treatments [5]. Taste signalling begins in specialized chemosensory taste receptor cells in the taste buds, which are located within specialized structures called fungiform papillae (FP) on the tongue [6]. The number and shape of FP vary greatly from person to person, which may impact taste sensitivity and preferences for different products (specific foods) and, consequently, the frequency of their consumption [7]. FP density is often reported to be higher in individuals who are classified as tasters of 6-n-propylthiouracil (PROP) than those not tasting PROP (PNT) [8]. The ability to taste bitter compounds such as PROP, which may be partly explained by genetic variation in TAS2R38 polymorphisms, correlates with increased perception of taste intensity and other sensory–oral sensations [9,10]. Nevertheless, it remains unclear whether PROP status is related to the taste scores measured by Taste Strips, which is a validated psychophysical taste test [11]. The advantage of this test is its ease of use, which is important when examining children [12,13,14]. Several factors can modulate the taste function in children; these need to be considered and require further research on taste strips. It is believed that the taste function improves with the age of the child, although results to date have varied. For example, sucrose intensity has been shown to increase from childhood to adulthood, however, sensitivity to bitter compounds remains similar in children and adults [15].

It is known that taste and smell are the two basic senses that allow assessing the quality of food consumed. Their interrelationship is emphasized by the fact that it has been estimated that 95% of taste disturbances are caused by a loss of smell, not a loss of taste [16]. Taste sensitivity and olfactory function have rarely been assessed together in the same study. Some previous reports advocated the statistical independence of chemosensory sensitivities [17].

The study aimed to describe sensory taste perception, measuring the number of fungiform papillae (FP) by the photo method and further examining whether the number of FP was associated with sensitivity to sweet, bitter, salty, and sour tastes as well as the olfactory function in children aged 11 to 15 years.

## 2. Materials and Methods

One hundred and one children aged 11–15 were qualified for the study. Children were recruited from a local school in Podkarpackie Province, Poland. The population was homogeneous in terms of ethnicity. The inclusion criteria for children were being 11–15 years of age and having parents’ consent to participate in the study. The exclusion criteria for children included the inability to test the taste and olfactory samples used in the study (e.g., taking any prescription drugs that may have interfered with their ability to taste, runny nose or other ailments that may have affected the olfactory function) and having an implanted pacemaker (contraindications for the bioimpedance test). Participants were asked to abstain from eating, drinking (except water at room temperature), brushing teeth, and chewing gum for 1 h before the test. The study was conducted following the ethical standards included in an appropriate version of the Declaration of Helsinki and Polish regulations. The study was approved by the institutional Bioethics Committee at the University of Rzeszow on 2 June 2015 (Resolution No. 15 June 2015) and by all appropriate administrative bodies.

### 2.1. Measuring Fungiform Papillae

The study used the method of measuring the FP number described by Shahbake et al. [18]. Before the start of the measurements, each subject rinsed their mouth with demineralized water. The examination was carried out in a sitting position so that the subject could put their elbows on the table and rest their chin on them. Then, the subject pulled out their tongue, which was dried with filter paper (Whatsmann No. 1). To colour the tongue, a circled blue filter paper (E133 brilliant blue) was placed on the left side of the centreline on the tip of the tongue with a pair of tweezers. After 5 s, the coloured filter paper was removed. Additionally, to match the photo with the participant, an individual sticky label with the identification number was attached to the child’s chin. At least three close-ups photos of the tongue were taken with a digital camera (Canon EOS 550D, Tokyo, Japan). All photos were transferred to a computer and processed and analysed using GIMP version 2 (GNU Image Manipulation Program for Windows). Each participant’s FP number was counted in duplicate by two independent investigators, and the mean FP number was calculated to obtain the FP number on a 6 mm diameter circle.

### 2.2. Assessment of Taste Function

Taste function was assessed by using Taste Strips (Burghart, Wedel, Germany). Taste Strips are filter-paper strips impregnated with a taste solution and determine sour, sweet, salty, and bitter taste scores [13]. Each time, one of four concentrations of sweet taste (0.05, 0.1, 0.2, and 0.4 g/mL sucrose), bitter taste (0.0004, 0.0009, 0.0024, and 0.006 g/mL quinine hydrochloride), sour taste (0.05, 0.09, 0.165, and 0.3 g/mL citric acid), or salty taste (0.016, 0.04, 0.1, and 0.25 g/mL sodium chloride) was used in the same pseudorandomized order, starting with the lowest concentration. Each participant was asked to abstain from eating and drinking (except water) 1 h before the examination. To assess a taste, strips were placed on the middle of their tongue for a maximum of 30 s. The participant then discarded the strip and took a sip of bottled water to rinse the water around the mouth. They were then asked to indicate if the strip taste was sour, sweet, salty, bitter, or tasteless. The score range for each taste quality was from 0 to 4. The total score was calculated as a sum of the scores of all of the performed tests (ranging from 0 to 16). A higher total score indicated a better taste function. PROP taster status was examined by a filter paper strip impregnated with PROP (Sensonics International, Haddon Heights, NJ, USA, 20 µg/strip). Each participant was asked to place the strip on the dorsal surface of their tongue for a maximum of 30 s. After this time, they were asked whether they recognized any taste (yes/no). Children who answered no or indicated that the strip tasted like paper were classified as nontasters. Children who mentioned that the strip taste was bitter, sour, bad, or spicy were classified as tasters. Moreover, participants who immediately removed the strip because of its foul taste or showed other signals of taste rejection were also classified as tasters [19].

### 2.3. Olfactory Testing

We used the 12-item U-Sniff odour identification (ID) test to initially assess the olfactory functions of the respondents. The ID test is used to assess the odour function in children aged 6 to 17 years. The test results can be used to differentiate among normosmia, hyposmia, and anosmia. The identification test includes 12 pens, each containing a familiar odour. The procedure is based on four alternative forced choices for each pen. During the test, the participant is asked to indicate one of four odours from a list related to the pen. When the selection is correct, 1 point of score is assigned. The procedure is repeated for all 12 pens. The overall score range is from 0 to 12. Higher total scores indicate better olfactory performance [20,21,22].

### 2.4. Anthropometric Evaluation

Participants’ height measurements were made three times with a portable SEA 213 stadium, with an accuracy of 5 mm, in a standing position, wearing light fabrics and without footwear. In the analyses, the average figure of the three measurements was included. Body weight was assessed with an accuracy of 0.1 kg using a body composition analyser (BC-420, Tanita, Tokyo, Japan). Body mass index (BMI) was calculated as weight (kg)/height (m)^2^ based on BMI values. The BMI percentiles of individual children were calculated based on BMI percentile charts specific for age, sex, and body height. Percentile charts that were developed under the Polish project entitled “Developing standards of blood pressure in children and adolescents in Poland”, OLAF, were used [23]. Based on the BMI percentile values, underweight (<5th percentile), normal weight (between 5th and 85th percentile), overweight (BMI ≥85th percentile and <95th percentile), and obese (≥95th percentile) statuses were determined.

### 2.5. Statistical Analysis

Descriptive statistics were calculated: the mean, standard deviation, median, and first and third quartiles (Q1–Q3). Compliance with the normal distribution of quantitative variables was checked using the Shapiro–Wilk test. To check the differences between two groups (boys and girls, tasters and nontasters, bitter taste score groups) for the analysed quantitative or ordinal variables, the Mann–Whitney U test was used. Spearman correlations were used to test the relationships between quantitative variables. Statistical analyses were performed using PS IMAGO PRO 7 (IBM SPSS Statistics 27, Armonk, NY, USA). The level of statistical significance was set at *p* < 0.05.

## 3. Results

### 3.1. The Number of Fungiform Papillae

The number of FP was analysed in 101 children (46 girls and 55 boys; mean age 13.0). The characteristics of this sample are indicated in Table 1. The median of the numbers of FP within the defined circle was 27 (interquartile range: 22–35); the range was from 12 to 47. Of the children, 6.9% had more than 40 FP. No differences were identified between girls and boys regarding the number of FP (Table 2).

Of the children, 69.2% (*n* = 70) had normal development according to the percentile grids, 13.9% (*n* = 14) were overweight, and 11.9% (*n* = 12) were obese. There were also no differences in the number of FP in relation to children’s body weight (normal/overweight or obese). The distribution of the numbers of FP is presented in Figure 1. We did not observe a clear tendency of increasing numbers of FP with the age of the respondents.

We also noticed differences in FP scores between tasters and nontasters (*p =* 0.0211). The scores were higher in the group of tasters, for which the median score was 31 (23.0–37.0), than in the group of nontasters, for which the median score was 25 (22.0–31.0) (Figure 2).

### 3.2. Taste and Smell Sensitivity

The median total taste score (sensitivity to all tastes) was 13.0 (11.0–14.0). The total scores were higher in the group of girls, for which the median score was 14 (12.0–15.0), than in the group of boys, for which the median score was 12 (9.0–13.0) (Table 2). However, there were no significant differences between the sexes in the individual taste tests (sweet, bitter, salty, sour). Median scores for the individual taste qualities were as follows: sweet, 3.0 (3.0–4.0); sour, 3.0 (2.0–3.0); salty, 4.0 (2.0–4.0); and bitter, 4.0 (2.0–4.0). About 30% of the examined children had the highest results in the assessment of sweet, bitter, and salty tastes. In the case of the sour taste, only 4% of the children rated all four stripes with the sour taste correctly. In total, 52 children (51.5%) were PROP tasters, while 32 (31.7%) participants reported feeling no sensation or reported that the strip tasted like paper. Another 17 children (16.8%) claimed that they felt something but could not qualify or define their sensations correctly. Most of the children, *n* = 87 (86.1%), showed normative results in the odour identification test. This means that their scores were equal to or greater than 10. Fourteen children had scores below 10, which indicated hyposmia, a partial loss of the sense of smell [24]. We found an association between the taste sensitivity to bitter taste and the number of FP (*p* = 0.0001). The distribution of the numbers of FP in categories of bitter taste sensitivity is presented in Figure 3. The obtained results clearly showed that children who correctly recognized the bitter taste (score = 4) were characterized by higher numbers of FP. The median score for children who were the most sensitive to bitter taste (score = 4) was 34.0 (27.0–37.0). For those who had some problems with correctly assessing all bitter taste samples (score = 0–3), the median score was 24.0 (20.0–31.0).

Spearman correlation analysis also revealed a weak association between the taste sensitivity to all tastes and the FP number (*r =* 0.30, *p* = 0.01). There was no association between the number of FP and taste sensitivity for sweet, salty, or sour. PROP status was associated with bitter taste score and total taste score. In addition, we found a positive correlation between bitter taste score and total taste score (Table 3).

## 4. Discussion

The number of FP was measured in 101 children between 11 and 15 years of age by a method that has been used in the past. The median number of FP of this sample was 29. This value was similar to the results of previous studies [18,25]. According to Correa et al. [26], FP numbers stabilize at the ages of 9–10, while the distribution and growth of fungiform papillae stabilize at 11–12 years of age.. Our results confirmed that the number of FP did not increase in children aged 11–15. In the age group 8–11, a decrease in the number of FP was demonstrated with age [25]. Our result confirmed earlier reports of higher numbers of FP in children than in adults [27]. However, it is believed that in adulthood, the number of FP falls with age [28]. We found no differences in the number of FP between boys and girls. Correa et al. [26] also did not show differences in the number of FP between male and female participants. Earlier reports on the relationship between sex and the density of FP in adults have shown conflicting findings [29,30]. Karikkineth et al. reported that women had a higher fungiform papillae density than men [31]. In our study, we could not confirm a relationship between BMI and the number of FP. There was also no association between body composition (fat content %) and the number of FP. Previous findings concerning the relationship between taste sensitivity and BMI in a young sample have been inconsistent [32,33,34,35]. Herz et al. [11] suggested that the differences in sensitivity to taste between people with high and normal BMI may vary depending on the stimulus of the trigeminal nerve and the nonbearing stimulus, of which the taste stripes are not assessed. It is worth noting that our taste test results were consistent with the work of van den Brink et al. [36], which provided the normative values of Taste Strips for children. The lowest test values were achieved for the sour taste. The combined result of the taste tests’ median total scores was also the same at 13.0 (11.0–14.0). Sex differences in taste function among children were reported by James et al. [37] based on freshly prepared taste solutions tests. During a study on children aged 8–9, he proved higher detection thresholds for sucrose and sodium chloride in boys than in girls. In a recent study of a large group of children using the Taste Stripes tool, girls outperformed boys, showing a combined total taste score significantly higher than that of boys as well as higher sweet and sour scores [36]. A similar trend was discovered in the current study, where girls were able to detect all four taste qualities better than boys. The girls’ scores were generally higher, but no differences were found in the individual taste qualities (sweet, salty, sour, and bitter). This may indicate, as suggested by other authors, that the taste function matures slightly faster in girls [12]. Several studies have been carried out previously on the relation between the taste state of PROP, the density of FP, and responses to oral stimulation in adults, but these relationships remain unobvious. Some studies indicated that the thresholds for detection and recognition of sweet, sour, salty, bitter, and umami tastes correlated neither with the bitterness of PROP nor with the density of FP. Only the detection and recognition thresholds were related to each other, which emphasizes the complexity of identifying the taste function [38,39]. In a study by Dinnella et al. [40] conducted on a large group of respondents, PROP bitterness ratings were positively correlated to the intensity of bitterness, sourness, sweetness, umami, and pungency, whereas no significant correlations were found between the density of FP and any taste or oral sensation intensity classification. In our study, PROP status was related to taste function in children, with higher total taste scores in tasters than in nontasters (*r* = 0.374; *p* = 0.01). It can therefore be concluded that PROP status is related to the taste function, but not so much that it can be seen as a measure of the general function of taste in children. We also analysed the association between the number of FP and the results of the taste function. In a study by JiIani et al. [25] on a group of children aged 8–11 years, no relationship was found between the number of FP and the threshold of bitter taste. In our study among older children, there was a tendency to higher results, which means that more sensitivity to bitter taste was observed in children with more fungiform papillae. This was the most clearly noticed in children who recognized the bitter taste even at the lowest concentration of the bitter taste pattern. However, no relationships were found between the number of FP and sweet, salty, or sour taste. Not all results of the studies performed so far have confirmed the relationship between the quantity of FP and the perceived intensity of bitterness. It seems that several environmental factors may diminish the FP response to oral stimuli and weaken the direct association between the density of FP and perceived intensity of bitter taste [41]. We also found higher numbers of FP in tasters compared to nontasters (*p* = 0.0211). Several recent studies failed to prove an association between FP count and PROP rating [29,40,42], whereas other studies identified significant associations [43,44,45]. Some of them confirmed the lack of a simple linear relationship between the PROP phenotype and the density of FP, both in the entire population and in selected age- and sex-matched samples. It was shown that the correlation of the number of FP with PROP intensity was dependent on the location on the tongue measured [46]. There are many other factors, methodological, genetic, demographic, and environmental, that may explain the current uncertainty of the relations between the number of FP and taste responses.

In our study, we also evaluated the olfactory function of our respondents. Several tests are available to examine olfactory function in children. The odour identification abilities in a paediatric population are well described and understood [47]. Based on the current literature, the U-Sniff odour identification test a tool that can be recommended for a correct and reliable olfactory test in children. As in a study by Gellrich [21], girls outperformed boys in U-Sniff odour identification (ID) test. In that study, using the 10th percentile to distinguish normosmia from reduced smell, in the age groups of 12–14 and 15–17 years, 10 points for (ID) test were obtained. This coincided with our observations as well as the guidelines of the manufacturer, U-Sniff. On the basis of the test results, hyposmia was found in 13.9% of children. In a large study on a population of ages 11–20 in which a full procedure was carried out, covering three tests (comprising threshold (T), discrimination (D), and identification (I) subtests), hyposmia was found in 19.5% of respondents [48]. In the current study, children scored a mean ± SD of 10.76 ± 1.13 points out of 12. The mean scores were higher than results reported by Schriever et al. (mean ± SD of 9.88 ± 1.80) [24], but this is explainable by the older ages of the children. The youngest age group (6–8 years) scored lower than three other age groups on the U-Sniff odour identification test [21]. The results of the taste tests were often compared with the number/density of FP, giving different conclusions. We found a link between the number of FP and U-Sniff test scores. Children with higher numbers of FP obtained higher results on the test assessing their olfactory functions. Although this relationship is interesting, there is no scientific explanation for it; therefore, we conclude that it may be coincidental and should not be taken into consideration. A significant correlation was also found between the results of the U-Sniff test and the results of the bitter taste test (*r* = 0.351; *p* < 0.001). Certain odours and certain tastes appear to share common perceptual properties. It was previously found that the results of the sweet taste test turned out to be correlated with the outcomes from another test performed in the same study on evaluating the similarity between sweet smells and sucrose and the degree of sweetness of these odours [49]. A study Stevenson et al. [50] demonstrated that the effects of odours on taste perception were not limited to sweetness enhancement and applied to sour as well as sweet tastes. Other studies have also confirmed the interdependence of taste and olfactory sensations. Yeoman’s data suggested that individual differences in the assessment of saccharin reliably accounted for subsequent results on evaluations of odours combined with saccharin. It is also worth mentioning that sensory changes reflected the differences in sensory quality between PROP taster and nontaster groups [51]. The overlapping of taste and olfactory sensations during sensory tests has been frequent, but this has not always implied a simple, unchanging relationship. This was confirmed in the latest work by Lim et al. [52], which demonstrated the independence of taste and smell in young and middle-aged subjects despite their overlapping during sensory experiments.

## 5. Conclusions

In conclusion, the current study assessed at the same time the taste and olfactory functions of children aged 11–15 using previously used and recognized methods. Our results showed an association between the number of FP and sensitivity to bitter taste but no association in the cases of sweet, salty and sour tastes. We also confirmed some differences in the numbers of FP between PROP tasters and nontasters. A limitation in our results was the small number of children tested. In addition, the identified dependencies did not correlate strongly with each other, so they should be treated with some caution. More detailed checking is required in the case of positive correlation between sensitivity to bitter taste and olfactory function. In future studies involving children, the assessment of taste and olfactory function should be performed together, as a single test cannot capture the entirety of the sensory experience.

## Figures and Tables

**Figure 1 nutrients-14-02578-f001:**
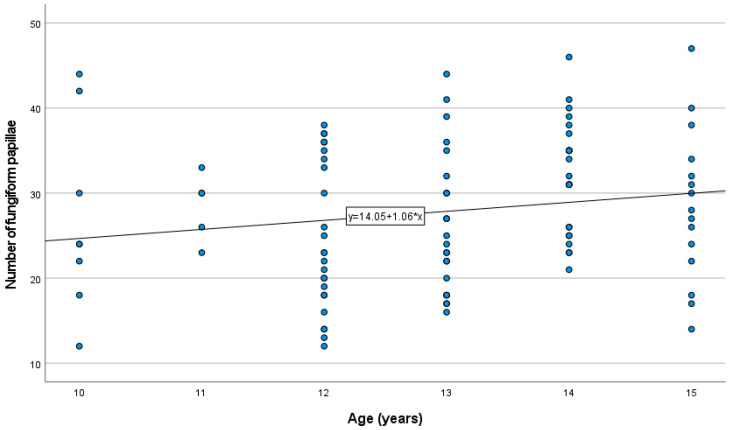
The distribution of the numbers of FP in categories of age (years).

**Figure 2 nutrients-14-02578-f002:**
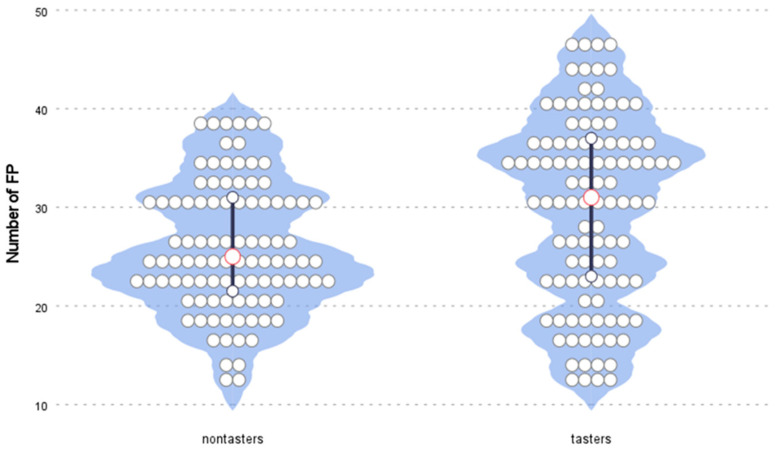
The distribution of the numbers of FP in the categories of tasters/nontasters.

**Figure 3 nutrients-14-02578-f003:**
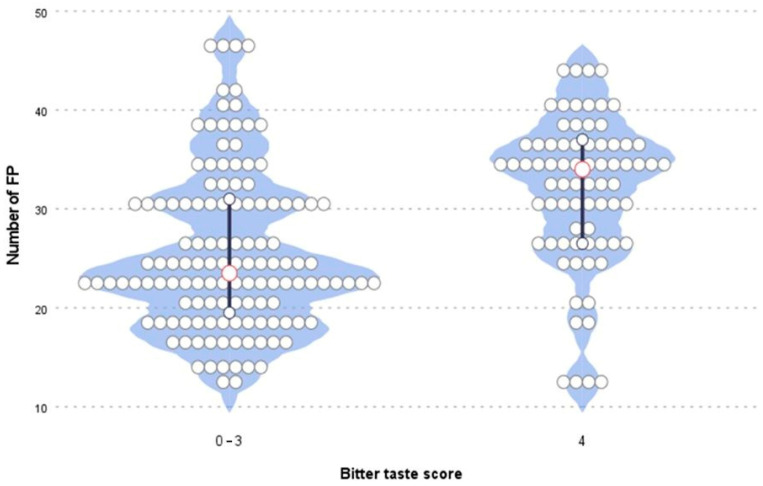
The distribution of the numbers of FP in categories of the bitter taste sensitivity.

**Table 1 nutrients-14-02578-t001:** Characteristics of the study sample (anthropometric data).

	All	Girls	Boys	*p*
N (%)	101 (100)	46 (45.5)	55 (54.5)	
Age (years) *	13 (12–14)	13 (12–14)	13 (12–14)	0.4061 ^a^
Height *	158 (149–156)	158.5 (148–166)	158 (150–165)	0.5550 ^a^
Weight *	47.4 (37.7–57.6)	27.8 (37.2–60.1)	48.2 (38.1–57.6)	0.5971 ^a^
BMI (percentile grids)				0.7563 ^b^
<5 percentile	5 (5.0)	2 (4.3)	3 (5.5)	
5–85 percentile	70 (69.2)	30 (65.3)	40 (72.7)	
>85 percentile	26 (25.8)	14 (30.4)	12 (21.8)	
Total fat content (%) *	15.4 (12.9–23.9)	20.4 (14.8–30.3)	12.9 (10.5–18.8)	<0.0001 ^a^

* Median with interquartile range. ^a^—Mann-Whitney U test. ^b^—χ^2^ test.

**Table 2 nutrients-14-02578-t002:** Characteristics of the study sample (performed tests).

	All	Girls	Boys	*p*
Number of fungiform papillae *	27 (22–35)	27 (22–35)	27 (22–34)	0.6255 ^a^
PROP status				
Tasters (%)	49 (48.5)	19 (41.3)	30 (54.5)	0.1848 ^b^
Nontasters (%)	52 (51.5)	27 (58.7)	25 (45.5)	
Sensitivity to sweet taste *	3.0 (3.0–4.0)	3.0 (3.0–4.0)	3.0 (3.0–4.0)	0.3550 ^a^
Sensitivity to salty taste *	3.0 (2.0–4.0)	3.0 (2.0–4.0)	3.0 (2.0–4.0)	0.7831 ^a^
Sensitivity to sour taste *	3.0 (2.0–3.0)	3.0 (2.0–3.0)	3.0 (2.0–3.0)	0.6014 ^a^
Sensitivity to bitter taste *	3.0 (2.0–4.0)	3.0 (2.0–4.0)	3.0 (2.0–4.0)	0.1321 ^a^
Sensitivity to all taste *	13 (11–14)	14 (12–15)	12 (9–13)	<0.0001 ^a^
U-Sniff test (odour test) *	11 (10–12)	11 (10–12)	11 (10–12)	0.2455^a^

* Median with interquartile range. ^a^—Mann-Whitney U test. ^b^—χ^2^ test.

**Table 3 nutrients-14-02578-t003:** Matrix for Spearman correlation between variables: FP, PROP, all tastes, bitter taste (*n* = 101).

		FP	PROP	All Tastes	Bitter Taste
FP	r	1.000	0.210	0.300	0.351
	P		0.035	0.002	<0.001
PROP	R	0.210	1.000	0.374	0.533
	P	0.035		<0.001	<0.001
All taste	R	0.300	0.374	1.000	0.657
	P	0.002	<0.001		<0.001
Bitter taste	R	0.351	0.533	0.657	1.000
	P	<0.001	<0.001	<0.001	

## Data Availability

Not applicable.

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
