# Peer review of "The Number of Fungiform Papillae, Taste Sensitivity and Smell Functions of Children Aged 11–15"

_nutrients, 2022, doi:10.3390/nu14132578_

Round 1

Reviewer 1 Report

Sobek and Jagielski conducted a study to assess taste function in 101 children aged 11-15 and found an interesting association between the number of FP and sensitivity to the bitter taste. Overall, the study is well-written and adds good information to the existing literature. However, there are some major and minor issues I would suggest the authors to correct in order to improve the overall quality of their work. 

Major comments

Even though I appreciate the authors did a quick smell identification test to rule out any taste dysfunction related to a smell loss, the correlation they found between the smell test and the number of FP seems to be just a coincidence in my opinion and does not have a rationale behind.

Why should patients with a higher number of FP smell better? I would suggest the author to delete this assumption in the results/conclusion sections unless they find a strong physiological/scientifical explanation for that.

Minor comments: 

Introduction: line 50: PROP definition was already given. No need to specify twice. Please correct. 

Material and methods

- Line 78: I would remove Pregnancy amongst the exclusion criteria just because of the age of the population enrolled.

- Ethnicity has been shown to be a variable influencing both taste and smell. Was the population homogenous in terms of ethnicity background? If not, differences in ethnicity could have influenced the findings observed. Please double-check that and mention it in this section. Please see: Pendolino AL, Ottaviano G, Scarpa B, Cattelan A, Andrews JA, Andrews PJ. Characteristics of taste dysfunction in COVID-19 subjects coming from two different countries. J Neurovirol. 2021 Jun;27(3):482-485. doi: 10.1007/s13365-021-00942-8.

Results

- Lines 176-177:  what does ‘Me’ mean? It has not been specified earlier in the test. However, if that means “median”, I don’t think it is an appropriate (i.e. widely used) acronym for that. I would avoid. 

Discussions:

- Line 212: I would keep the limitations of the study at the end of the paper. 

- Line 214: I think there is an extra space before the word ‘our’. Similarly in line 248

- Line 225: please correct ‘suggests’ with ‘suggest’

- Line 244: please use the abbreviation FP for fungiform papillae to make it consistent in the test. 

- Line 265: did the author mean “however this difference was not statistically different.”?

- Line 276: the word ‘well-described’ seems to be more appropriate instead of ‘well-examined’ in this context. 

Reviewer 2 Report

Nutrients 1772569: The number of fungiform and taste sensitivity and smell functions children aged 11-15

This study is testing the possible correlation between the sense of taste and fungiform numbers. The goal is interesting, however the paper is not ready for publication at the current stage. First 

In the abstract, there are abbreviations without the full description. Please add the full words, and the abbreviations in parentheses if you want, in the abstract. Abstract become shown in Pubmed and others, and it is necessary to use full words or full words with abbreviation in parentheses. 

Line 109: how long was it placed? I see 30 sec in line 117. Was it the same in the experiment described in 109? Please write the time length here as well. Also, please describe the rationale to use 30 sec.

Line 145: “2.5. Statistical analysis” – The heading needs to be as same as 2.4, in italic, space between the previous section, left side margin. 

Are you intentionally using “,” and “.” mixed in writing like, for example 31,7% and 16.8%? This is very annoying and suggests lack of careful attention to details.

Why are you writing percentile as “percentile” in some places and “%tile” in others, even in the same one table. Like the “,” and “.”, it is very annoying and suggests lack of careful attention to details. Please be consistent. I recommend using “percentile”.

Line 191: “it is clearly visible that children who correctly recognized the bitter taste were characterized by a higher number of fungiform papillae”. -> Figure 2 does not convince me at all that tendency.

Figures: The linear regression is not very convincing. 
